# No-Regret Online Autobidding Algorithms in First-price Auctions*

**Yilin Li**
Chinese University of Hong Kong
ylli25@cse.cuhk.edu.hk

**Yuan Deng**
Google Research
dengyuan@google.com

**Wei Tang**
Chinese University of Hong Kong
weitang@cuhk.edu.hk

**Hanrui Zhang**
Chinese University of Hong Kong
hanrui@cse.cuhk.edu.hk

## Abstract

Automated bidding to optimize online advertising with various constraints, e.g. ROI constraints and budget constraints, is widely adopted by advertisers. A key challenge lies in designing algorithms for non-truthful mechanisms with ROI constraints. While prior work has addressed truthful auctions or non-truthful auctions with weaker benchmarks, this paper provides a significant improvement: We develop online bidding algorithms for repeated first-price auctions with ROI constraints, benchmarking against the optimal randomized strategy in hindsight. In the full feedback setting, where the maximum competing bid is observed, our algorithm achieves a near-optimal $\widetilde{O}(\sqrt{T})$ regret bound, and in the bandit feedback setting (where the bidder only observes whether the bidder wins each auction), our algorithm attains $\widetilde{O}(T^{3/4})$ regret bound.

## 1 Introduction

*Autobidding* has become the predominant paradigm in online advertising. The idea is that advertisers compete for ad opportunities through repeated auctions, where each advertiser employs an automated bidding algorithm (i.e., an autobidder) to bid on behalf of the advertiser. Autobidders are essentially online constrained optimizers: As auctions happen over time, an autobidder maximizes on the fly a certain objective quantity, e.g., clicks or conversions, subject to constraints, e.g., the total spending cannot exceed a predetermined budget, or the ratio between the return and the investment (i.e., the return-on-investment, or ROI ratio) must be at least a given target ratio. Normally, advertisers are free to customize these components of autobidders to better serve their own goals.

Over years, advertisers seem to have converged to the following common practice: An overwhelming majority of advertisers use autobidders that behave like *ROI-constrained value maximizers*. This highlights the importance of designing good online bidding algorithms in this very setting. The problem, roughly speaking, is the following: Initially, an auction mechanism (e.g., first-price or second-price) is chosen externally by the advertising platform, which is used in all subsequent auctions. At each step, an auction happens, and the algorithm observes the "value" of winning in this auction. Then, the algorithm must immediately submit a bid based on historical observations and the value, after which the algorithm observes the outcome of the auction, i.e., whether it has won and possibly also the highest bid by other advertisers. In the long run, the algorithm must (approximately) maximize the total value obtained over all auctions, subject to the constraint that the overall ROI ratio is at least a predetermined target ratio. The key challenge here is that at each step, the algorithm

---

*The full version of the paper is available at `https://arxiv.org/abs/2510.16869`.

39th Conference on Neural Information Processing Systems (NeurIPS 2025).

must decide the bid without knowing the future, or even the competing bid in the current auction. Nonetheless, it must ensure that its decisions are almost optimal from hindsight (i.e., it has *no regret*), and the ROI constraint is approximately satisfied over the entire horizon.

Indeed, considerable effort has been invested into this task. Here, we refrain from a prolonged discussion (see Section 1.1 for further related work) and focus on the most related results. Feng et al. [21] study the problem when the auction mechanism is *truthful*, and give an almost optimal no-regret online bidding algorithm. Despite its undoubted importance, one severe limitation of their result is that it does not apply to some of the most common auction mechanisms in reality, including first-price auctions and generalized second-price auctions — for these mechanisms are not truthful. To address this issue, Aggarwal et al. [3] design alternative no-regret bidding algorithms that work for nontruthful auctions. However, their algorithms only have no regret against a weaker benchmark, i.e., the optimal *Lipschitz-continuous* bidding strategy from hindsight, which in general can be much worse than the unconditionally optimal bidding strategy. This leaves the following question open: *Is it possible to design a no-regret online bidding algorithm against the unconditionally optimal strategy for nontruthful auction mechanisms?*

**Our results.** We provide an affirmative answer to the above question. For concreteness, we derive our results for first-price auctions (which is the most representative nontruthful auction mechanism). We consider two natural feedback models commonly studied in the literature:

- *Full feedback*: the algorithm learns the auction outcome and the competing bid, after each auction;
- *Bandit (or one-bit) feedback*: the algorithm learns only the outcome of the auction and nothing else after each auction.

The latter setting is more challenging than the former, since the algorithm obtains less information.

In the full feedback setting, a folklore lower bound of $\Omega(\sqrt{T})$ is essentially inherited from the most fundamental setting of stochastic bandits - Slivkins et al. [34] prove the a lower bound regret $\Omega(\sqrt{KT})$, where $K$ is the number of arms and $T$ is time horizon in stochastic bandits.[1] We present an online bidding algorithm that achieves the *optimal* (up to polylog factors) regret bound of $\widetilde{O}(\sqrt{T})$ against the *unconditional, possibly randomized, optimal strategy* in hindsight, where $T$ is the time horizon. This generalizes the result by Feng et al. [21] and strengthens the one by Aggarwal et al. [3] in the same setting. In the bandit feedback setting, we present an algorithm that achieves a regret bound of $\widetilde{O}(T^{3/4})$ against the unconditionally optimal strategy, which strengthens the result by Aggarwal et al. [3] in the same setting.[2] These results paint a considerably clearer picture of online bidding algorithms for ROI-constrained value maximizers.

**Technique overview.** The key technical challenge in designing online autobidding algorithms, which has also been observed by Aggarwal et al. [3], is that unconditionally optimal bidding strategies appear to have rich structures. This tends to baffle existing techniques, which normally require the class of strategies optimized over to be "simple" and / or "structured" in some way. Below we first briefly review how prior work deals with this challenge.

Feng et al. [21] restrict their attention to truthful auctions, in which optimal bidding strategies admit a clean characterization: Without loss of generality, they only need to consider "uniform bid scaling", i.e., bidding strategies defined by a single multiplier $\theta$, such that the strategy places a bid of $\theta \cdot v$ when the value of winning is $v$. Restricted to such bidding strategies, one essentially only needs to optimize the multiplier $\theta$, which allows Feng et al. [21] to essentially reduce their problem (in a nontrivial way) to one that can be handled within the powerful mirror descent framework. Aggarwal et al. [3] take an approach that is in a sense more general: They consider bidding strategies that are Lipschitz-continuous, which subsume uniform bid scaling as a subclass. By doing so, Aggarwal et al. [3] manage to further incorporate technical insights from the Lipschitz bandits literature, which help establish nontrivial regret bounds for nontruthful auctions.

We take an approach that is in spirit similar to [21]. We first make a more general structural observation, that even in nontruthful auctions, optimal (and possibly randomized) bidding strategies

---

[1] A detailed discussion about the lower bound is provided in the full version of the paper [16].

[2] In the bandit feedback setting, Aggarwal et al. [3] establish a lower bound of $\Omega(T^{2/3})$, closing the gap between this lower bound and our upper bound remains an important open direction for future work.

are *essentially parametrized by a single number*, which roughly corresponds to the marginal ROI of the bidding strategy. We present an algorithmically compatible characterization of this class of strategies (see Section 3), which allows us to (1) generalize the mirror-descent-based algorithm used in [21] to nontruthful auctions and (2) without loss of generality, focus on deterministic bidding strategies, provided one condition: The algorithm needs to *know the distribution of competing bids*.[3] This condition is crucial, since the structure of optimal bidding strategies necessarily depends on the distribution of competing bids, and without knowing the former, one simply cannot guarantee the conditions for the mirror-descent primitive to produce the desired regret bound.[4] In our setting, of course the algorithm cannot know this distribution beforehand. Below we discuss how this requirement can be removed.

We first show that the requirement can be *relaxed* without hurting the regret too much. In particular, we show that if we run the generalized mirror-descent-based algorithm with a slightly inaccurate distribution of competing bids given as input, then its per-round regret can only degrade to the extent that the input distribution is inaccurate. With that as a primitive, we construct an algorithm that "bootstraps" itself without any prior knowledge about the environment in the full feedback setting (see Section 4). The algorithm "restarts" itself periodically as it gathers more information about the distribution of competing bids. Upon each restart, it runs a fresh instance of the mirror-descent-based primitive with the latest estimated distribution as its input, which allows it to gradually lower the per-round regret over time as the estimation becomes more accurate. By properly pacing the restarts, we manage to beat the $O(T^{2/3})$ bound via simple exploration-exploitation and achieve an almost optimal bound of $\widetilde{O}(\sqrt{T})$.

In the bandit feedback setting, the above strategy does not work any more, and we have to fall back to simple exploration-exploitation. In our setting, even this conceptually simple high-level strategy requires nontrivial technical ingredients to implement: In fact, here we need a slightly stronger property of the primitive, because we can no longer guarantee a *uniform* estimation error of the distribution of competing bids. We show that the property holds for a slightly adapted version of the primitive used in the full feedback setting. With that as a building block, by optimizing the granularity of discretization and the tradeoff between exploration and exploitation, we manage to establish a regret bound of $\widetilde{O}(T^{3/4})$ (see Section 5), which matches the guarantee established by Aggarwal et al. [3] against the weaker Lipschitz-continuous benchmark. Notably, the ideal $O(\sqrt{T})$ bound is impossible in this setting due to a lower bound of $\Omega(T^{2/3})$ by Aggarwal et al. [3]. We leave closing this regret gap as an important furture direction.

## 1.1 Related Work

Most closely related to our work is the line of research on online bidding algorithms in repeated auctions, with ROI-constrained value-maximizing bidders. In the paradigm with ROI constrained value maximizing bidders, Feng et al. [21] propose an algorithm based on the primal-dual framework proposed by Balseiro et al. [6], which achieves a nearly optimal $\widetilde{O}(\sqrt{T})$ regret bound for truthful auctions. Castiglioni et al. [14] propose an algorithm that achieves no regret in repeated, possibly nontruthful, auctions, under the assumption that the spaces of values and bids are both finite. Aggarwal et al. [3] design an algorithm which works with nontruthful auctions and continuous value / bid space, and achieves no regret against any Lipschitz-continuous bidding strategy. Our work generalize / strengthen all these results by achieving the nearly optimal $\widetilde{O}(\sqrt{T})$ regret against the unconditionally optimal bidding strategy without further assumptions.[5] Also conceptually related is the work by Vijayan et al. [36], who consider a "harder" setting where the algorithm does not even observe the value of winning, and instead, must learn to estimate this value on the fly. Their result is orthogonal to ours, in particular because they also assume a finite space of bids.

Beyond autobidding, there is an extremely rich body of research on online bidding algorithms in repeated auctions. To name a few results: Balseiro et al. [6] propose a primal-dual framework and apply their framework to repeated second-price auctions with budget constraints and derive no-regret bidding algorithms. Wang et al. [37] study the same problem in repeated first-price auctions.

---

[3]Note that the problem remains nontrivial, because the algorithm does not know the distribution of its value.

[4]This is not an issue in [21] since the structure they need is independent of the distribution of competing bids.

[5]The $\widetilde{O}(\sqrt{T})$ bound applies to the full feedback setting. In the bandit feedback setting, we match the bound by Aggarwal et al. [3] against a weaker benchmark, thereby also strengthening their result.

Cesa-Bianchi et al. [15] study online bidding algorithms in first-price auctions without budget or ROI constraints, and pin down the optimal regret in a number of settings. Kumar et al. [28] present no-regret bidding algorithms that are strategically robust. Susan et al. [35] design no-regret bidding algorithms for multi-platform settings, where the bidding algorithm repeatedly participates in multiple parallel auctions. Berriaud et al. [11] discuss the repeated auction problem in artificial currency mechanisms with payment redistributed uniformly in each round. More generally, there is a long line of research on no-regret algorithms and dynamics for allocation problems under constraints, e.g., [5, 9, 12, 13, 22, 23, 24, 25, 26, 27, 31, 33, 38]. We refrain from an extensive discussion since these results concern fairly different problems from ours, both conceptually and technically.

Our results are also conceptually related to the growing literature on the design and analysis of auctions and marketplaces with autobidders. Prior research in this direction concerns various topics, including the efficiency of traditional auction mechanisms [2, 17, 18, 20, 29], the design of (approximately) optimal mechanisms [4, 8, 10, 19, 30, 32], etc. See the recent survey by Aggarwal et al. [1] for a comprehensive exposition.

## 2 Preliminaries

We consider an online bidding setting where a single bidder participates in single-item first-price auctions, repeated in $T$ rounds. At each time step $t \in [T]$, the learner realizes the value $v_t$ in the current round and then submits a bid $b_t$ based on the current value and the historical information up to now. The bidder wins the item if her submitted bid is not less than the (highest) competing bid $d_t$ ($d_t \in [0, 1]$) in this round, and we denote by $x_t(b_t, d_t) = \mathbf{1}\{b_t \geq d_t\}$ the ex post allocation outcome of the current round. If the bidder wins the item, she pays a price $b_t$ which equals to her submitted bid, otherwise she pays $0$. Therefore, the payment function, denoted by $p_t(b_t, d_t)$, is $p_t(b_t, d_t) = b_t \cdot \mathbf{1}\{b_t \geq d_t\}$. We focus on value-maximizing bidder where the bidder's utility at time $t$ is given by $r_t(b_t, d_t) = v_t \cdot x_t(b_t, d_t)$.

**Benchmark and regret definitions.** We assume that the sequence of value $v_t$ is drawn independently and identically (i.i.d.) from an unknown distribution, whose CDF is denoted as $H$. The bidder's objective is to design an online bidding algorithm ALG that maximizes her total realized utility subject to a Return-On-Investment (ROI) constraint.[6] Namely, her total utility must be at least $\rho$ times her total payment: $\rho \cdot \sum_{t \in [T]} p_t(b_t, d_t) \leq \sum_{t \in [T]} v_t \cdot x_t(b_t, d_t)$ where $\rho > 0$ is the target ROI ratio of the bidder.

$$
\begin{aligned}
\max \quad & \mathbb{E}\left[\sum_{t \in [T]} v_t \cdot x_t(b_t, d_t)\right] \\
\text{s.t.} \quad & \mathbb{E}\left[\rho \cdot \sum_{t \in [T]} p_t(b_t, d_t)\right] \leq \mathbb{E}\left[\sum_{t \in [T]} v_t \cdot x_t(b_t, d_t)\right] ,
\end{aligned}
\tag{$\mathcal{P}$}
$$

where the expectation is over the possible randomness of the bid sequence $\{b_t\}$ and the randomness of the maximum competing bid $\{d_t\}$. W.l.o.g., we assume that $\rho = 1$.[7]

We assume that the (maximum) competing bid $d_t$ at each round is i.i.d. realized from an unknown competing bid distribution $F$. In this context, the allocation and the bidder's payment in round $t$ are determined by her submitted bid $b_t$: the allocation function $x_t(b_t, d_t) = F(b_t)$ and the payment function $p_t(b_t, d_t) = b_t \cdot F(b_t)$.

We denote the bidder's value sequence by $V^T = (v_t)_{t \in [T]}$. An online bidding algorithm takes input of the bidder's historical information and the realized value at the current round, and outputs a (possibly randomized) bid. Given an online algorithm ALG, let $\mathrm{Reward}(\mathsf{ALG} \mid (F, V^T))$ (resp. $\mathrm{Payment}(\mathsf{ALG} \mid (F, V^T)))$ denote the bidder's total expected utility (resp. total expected payment)

---

[6]For pedagogical reasons and for presentation simplicity, we focus exclusively on ROI constraints. Our results and analysis can be readily generalized to the setting with additional budget constraints, e.g., through techniques introduced in [7]. The extension to budget constraints is discussed in the full version of the paper [16], in which we also present extensions to general non-truthful auctions.

[7]For $\rho \neq 1$, we can rescale bidder's value to be $v_t \cdot \rho$, and all our analysis/results can be carried over similarly.

given the value sequence $V^T$ and competing bid distribution $F$:

$$\text{Reward}(\mathsf{ALG} \mid (F, V^T)) = \mathbb{E}_{(b_t) \sim \mathsf{ALG}}\left[\sum\nolimits_{t \in [T]} v_t \cdot F(b_t)\right] \; ;$$

$$\text{Payment}(\mathsf{ALG} \mid (F, V^T)) = \mathbb{E}_{(b_t) \sim \mathsf{ALG}}\left[\sum\nolimits_{t \in [T]} b_t \cdot F(b_t)\right] \; .$$

Thus, the ROI violation is given by $\Delta(\mathsf{ALG} \mid (F, V^T)) = \text{Payment}(\mathsf{ALG} \mid (F, V^T)) - \text{Reward}(\mathsf{ALG} \mid (F, V^T))$.

Let $\mathsf{OPT}$ be the corresponding optimal bidding algorithm in hindsight for the program $\mathcal{P}$, i.e., $\mathsf{OPT}$ has knowledge about the competing bid distribution $F$ and the realized value sequence $V^T$. Thus, an algorithm's regret relative to $\mathsf{OPT}$ under the input value sequence $V^T$ is

$$\text{Regret}(\mathsf{ALG} \mid (F, V^T)) = \text{Reward}(\mathsf{OPT} \mid (F, V^T)) - \text{Reward}(\mathsf{ALG} \mid (F, V^T)) \; .$$

We measure the performance of an algorithm $\mathsf{ALG}$ by its expected regret over the distributions $F$ and $H$, defined as follows:

$$\text{Regret}(\mathsf{ALG} \mid (F, H)) = \mathbb{E}_{V^T \sim H^T}\left[\text{Regret}(\mathsf{ALG} \mid (F, V^T))\right] \; . \tag{1}$$

and expected ROI violation over the distribution $F$ and $H$:

$$\Delta(\mathsf{ALG} \mid (F, H)) = \mathbb{E}_{V^T \sim H^T}\left[\Delta(\mathsf{ALG} \mid (F, V^T))\right] \; .$$

**The feedback models.** We now specify the feedback that the bidder receives at the end of each round. In this paper, we consider the following two feedback models.

- **Full feedback:** At the end of round $t$, the bidder observes the competing bid $d_t$.
- **Bandit feedback:** At the end of round $t$, the bidder only receives a binary signal $\text{win}_t \in \{0, 1\}$ indicating whether she won the item or not, i.e., $\text{win}_t = \mathbf{1}\{d_t \le b_t\}$.

## 3  Optimal Randomized Autobidding Strategy

In this section, we first present the bidder's optimal bidding strategy in hindsight, where the sequence of values $V^T$ is known. We then design an online bidding algorithm that has low regret and low ROI violation when competing bid distribution $F$ is known, while value distribution $H$ remains unknown.

We first define the following allocation-payment curve, which will be helpful for our analysis.

**Definition 3.1** (Allocation-payment curve $G$). *Given a competing bid distribution $F \in \Delta([0, 1])$, an allocation-payment curve $G : [0, 1] \to [0, 1]$ is defined as: $G(b \cdot F(b)) = F(b)$ for all $b \in [0, 1]$.*

The allocation-payment curve $G$ captures the relationship between the payment and the allocation (as well as the reward, which is proportional to the allocation when the value of winning is fixed) induced by any bid $b$.

### 3.1  Characterizing Optimal Bidding Strategy

We start with the following example which shows the suboptimality of deterministic bidding strategies.

**Example 3.2** (Suboptimality of deterministic strategy). *Consider following competing bid distribution: $Pr[d = 0] = Pr[d = 1] = 1/2$. The bidder always has a value $v = 1/2$. Let $T = 1$, and the target ROI ratio $\rho = 1$. One optimal deterministic bidding strategy would be to always submit a bid $b = 0$, which yields an expected reward $1/4$ and an expected payment $0$ — note that restricted to deterministic bidding, there is no way to utilize this "slackness" in the ROI constraint.[8] On the other hand, consider the following randomized bidding strategy: $Pr[b = 0] = 2/3, Pr[b = 1] = 1/3$. This strategy yields an expected reward $1/3$ and an expected payment $1/3$. Both strategies satisfy the ROI constraint, while the randomized strategy gives a strictly higher reward.*

---

[8]Here and in the rest of the paper, we assume ties are broken in favor of the bidding algorithm. This makes no difference when the distribution of competing bids is continuous.

Example 3.2 shows that to maximize the bidder's expected reward, it may be necessary to solve program $\mathcal{P}$ over all randomized bidding strategies. This is (at least superficially) incompatible with the primal-dual framework that underlies all previous results on online autobidding, which presents new technical challenges. To address this issue, we present a constructive reduction that brings us back to the deterministic world, where we can cast the primal-dual framework more naturally. In particular, we show that given any input value sequence $V^T$ and any competing bid distribution $F$, it is always possible to construct another distribution $F_{\mathrm{conv}}$ that captures essentially the same tradeoff between reward and payment as $F$ induces, with one key difference: With $F$ replaced by $F_{\mathrm{conv}}$, we can focus our attention to deterministic bidding strategies without loss of generality.

**Theorem 3.1** (Optimal randomized bidding strategy). *Fixing any input value sequence $V^T$. For any competing bid distribution $F$, there exists a distribution $F_{\mathrm{conv}} \in \Delta([0,1])$ such that: For any (randomized) strategy under $V^T$ and $F$, there is a deterministic strategy under $V^T$ and $F_{\mathrm{conv}}$ inducing reward and payment that are both no worse, and vice versa.*

To prove Theorem 3.1, we first characterize the relation of allocation $F(b)$ (i.e. the probability of winning) and payment $F(b) \cdot b$ by the allocation-payment curve $G$, where $(F(b) \cdot b, F(b))$ is a point on the curve. Curve $G$ is parametrized by a deterministic bid $b$. To incorporate randomized bidding, we observe that randomizing between two bids corresponds to taking a convex combination of two points on the allocation-payment curve. As a result, the best tradeoff between allocation and payment under randomized bidding is given by the concave envelope $G_{conv}$ of the allocation-payment curve $G$. Therefore, if there is a distribution $F_{conv}$ whose corresponding allocation-payment curve is $G_{conv}$, the deterministic bidding strategies on $F_{conv}$ are equivalent to randomized bidding strategies on $F$ (i.e. for any randomized strategy on $F$, we can find a corresponding deterministic solution on $F_{\mathrm{conv}}$ and vice versa ). We prove the existence of $F_{conv}$ by constructing it from $F$.

With Theorem 3.1, we can simplify our algorithm design as follows: instead of designing an online bidding algorithm that has low regret against the optimal randomized bidding strategy under $(F, V^T)$, we can turn to designing an algorithm that has low regret against the optimal deterministic strategy under the corresponding $(F_{\mathrm{conv}}, V^T)$.

We conclude this subsection with a proof sketch of Theorem 3.1, where its full proof is quite technical, and thus, we defer it to the full version of the paper [16].

*Proof of Theorem 3.1.* Fixing any input value sequence $V^T$. For any competing bid distribution $F$, the proof of Theorem 3.1 consists of three main steps:

- **Step 1 – Prove that the "concave envelope" $G_{\mathrm{conv}}$ [9] of allocation-payment curve $G$ attains the maximum expected reward**: In our first step, we show that given any fixed value $v$, any (possibly randomized) bidding strategy that leads to an expected payment of $x$ has an expected reward at most $v \cdot G_{\mathrm{conv}}(x)$.
- **Step 2 – Construct the distribution $F_{\mathrm{conv}}$**: In this step, we construct such distribution $F_{\mathrm{conv}}$ that corresponds to an allocation-payment curve $G_{\mathrm{conv}}$, which is essentially the concave envelope of the allocation-payment curve $G$ induced by $F$.
- **Step 3 – Prove the reward equivalence**: In this step, we show that any reasonable randomized bidding strategy under $(F, V^T)$ is equivalent to generating a bid on the allocation-payment curve $G_{\mathrm{conv}}$, which corresponds to a deterministic bidding strategy under $(F_{\mathrm{conv}}, V^T)$, and vice versa. $\square$

### 3.2 No-Regret Autobidding Algorithm for Known Competing Bid Distribution

In this subsection, we describe a no-regret online bidding algorithm (see Algorithm 1) when the competing bid distribution $F$ is known to the bidder, while the value distribution $H$ still remains unknown. This algorithm will serve as a building block for our algorithm design when the both $F$ and $H$ are unknown to the bidder.

---

[9]Technically, $G_{\mathrm{conv}}$ might differ from the actual concave envelope of $G$, but this can happen only on the part of $G_{\mathrm{conv}}$ that never matters for our purposes.

We first note that after introducing the Lagrangian multiplier $\lambda \geq 0$, we can reformulate the program $\mathcal{P}$ as follows

$$\max \ \mathbb{E}\left[\sum_{t \in [T]} v_t \cdot F(b_t) + \min_{\lambda \geq 0} \lambda \cdot \sum_{t \in [T]} (v_t \cdot F(b_t) - b_t \cdot F(b_t))\right] \ . \tag{2}$$

Following a similar dual-prime framework proposed by [6], we proceed by choosing a bid $b_t$ at round $t$ to maximize the objective in Eqn. (2) using a fixed Lagrangian multiplier $\lambda_t$ and then update $\lambda_t$ to $\lambda_{t+1}$ use the observed feedback. Notice that given a fixed $\lambda_t$ at round $t$, the problem in Eqn. (2) is equivalent to

$$\max \ \mathbb{E}\left[\frac{1 + \lambda_t}{\lambda_t} \cdot v_t \cdot F(b) - b \cdot F(b)\right] \ . \tag{3}$$

As we show in Example 3.2, the optimal bidding strategy may be randomized. Thus, using Theorem 3.1, instead of directly solving the problem Eqn. (3), we consider solving a bid $\widetilde{b}_t$ that maximizes $\frac{1 + \lambda_t}{\lambda_t} \cdot v_t \cdot F_{\mathrm{conv}}(b) - b \cdot F_{\mathrm{conv}}(b)$. We then construct the equivalent randomized bidding strategy to be implemented in practice. We then update the dual variable $\lambda_t$ using the a generalized negative entropy in the mirror descent framework to adjust the penalty for the future rounds. We summarize the algorithm details in Algorithm 1.

$$\widetilde{b}_t \leftarrow \mathrm{argmax}_{b \in [0,1]} \left[(1 + \lambda_t) \cdot v_t \cdot F_{\mathrm{conv}}(b) - \lambda_t \cdot b \cdot F_{\mathrm{conv}}(b)\right] \ ; \tag{4}$$

---

**Algorithm 1** No-regret bidding algorithm for known $F$

---

**Input:** Time horizon $T$, competing bid distribution $F$, the value sequence $V^T$ (revealed sequentially).
**Initialize:** Initial dual variable $\lambda_1 = 1$ and the dual mirror descent step size $\alpha = 1/\sqrt{T}$
 1: Let $F_{\mathrm{conv}}$ be the distribution defined as in Theorem 3.1 (its construction is in the full version [16]).
 2: **for** $t \leftarrow 1$ to $T$ **do**
 3:     Observe the value $v_t$.
 4:     Compute $\widetilde{b}_t \leftarrow \mathrm{argmax}_{b \in [0,1]} \left[\frac{1+\lambda_t}{\lambda_t} \cdot v_t \cdot F_{\mathrm{conv}}(b) - b \cdot F_{\mathrm{conv}}(b)\right]$
 5:     Given $\widetilde{b}_t$, construct the equivalent randomized bidding strategy mentioned in Theorem 3.1 and realizes a bid $b_t$. /* Such randomized bidding strategy can be constructed efficiently, see the proofs in the full version [16]. */
 6:     Submit the bid $b_t$ (to the outer environment or algorithm).
 7:     Update $g_t(\widetilde{b}_t) = v_t \cdot F_{\mathrm{conv}}(\widetilde{b}_t) - \widetilde{b}_t \cdot F_{\mathrm{conv}}(\widetilde{b}_t)$.
 8:     Update $\lambda_{t+1} = \lambda_t \cdot \exp[-\alpha \cdot g_t(\widetilde{b}_t)]$.
 9: **end for**
10: **return** The bid sequence $\{b_t\}_{t \in [T]}$.

---

The performance of Algorithm 1 is detailed below, whose analysis follows similarly to [6, 21].

**Proposition 3.2.** *For any unknown value distribution $H$, the expected regret of Algorithm 1 under any competing bid distribution $F$ is bounded by* $\mathrm{Regret}($*Algorithm 1* $\mid (F, H)) = O(\sqrt{T})$*, and the expected ROI violation is bounded by* $\Delta($*Algorithm 1* $\mid (F, H)) \leq 2 \log T \sqrt{T}$*.*

## 4   No-Regret Autobidding Algorithm with Full Feedback

In Section 3, we present an algorithm (see Algorithm 1) when the competing bid distribution is known that achieves low expected regret and ROI violation. In practice, both the competing bid distribution $F$ and the value distribution $H$ may be unknown, and the bidder needs to use the observed feedback to learn these uncertainties by exploring different bids. In this section, we consider a full feedback model where the bidder can observe the realized competing bid $d_t$ at the end of each round, and we provide an algorithm (see Algorithm 2) with low expected regret and low ROI violation.

**Algorithm descriptions.** At a high-level, Algorithm 2 operates as a stage-based algorithm. In each stage $m \in [M]$ where $M$ is the number of stages determined shortly, we use the competing bids observed in all previous stages to maintain an empirical CDF $\widehat{F}_m$ for the underlying competing bid distribution $F$. Let $T_m \in \mathbb{N}^+$ be the number of rounds in stage $m$ and $N_m = \sum_{m' \in [m-1]} T_{m'}$ be the number of total rounds right before the stage $m$. We maintain $\widehat{F}_m$ as follows:

$$\widehat{F}_m(b) = \frac{1}{N_m} \sum_{i \in [N_m]} \mathbf{1}\{b \geq d_i\}, \quad b \in [0, 1], \tag{5}$$

To account for the estimation error and maintain optimism in our bidding strategy, we further define the optimistic CDF $\bar{F}_m$ as

$$\bar{F}_m(b) = \left(\widehat{F}_m(b) + \varepsilon_m\right) \wedge 1, \quad b \in [0, 1]. \tag{6}$$

where $\varepsilon_m$ is a parameter carefully chosen to balance the exploration and exploitation tradeoff.[10] We then implement Algorithm 1 by feeding the distribution $\bar{F}$ (while the actual competing bids are still generated according to the true unknown competing bid distribution $F$). We summarize the algorithm details in Algorithm 2.

---

**Algorithm 2** No-regret bidding algorithm with full feedback

---

**Input:** Time horizon $T$, the value sequence $V^T$ (revealed sequentially).
**Input:** Let $M = \lceil \log(T + 1) \rceil$ be the number of stages.
**Initialize:** Let $\bar{F}_1$ be an arbitrary distribution. $N_1 \leftarrow 0$
 1: **for** $m \leftarrow 1$ to $M$ **do**
 2:      Current stage length: $T_m \leftarrow 2^{m-1}$.
 3:      Feed the Algorithm 1 with inputs: $T \leftarrow T_m, F \leftarrow \bar{F}_m$. `/* Note that we compute the submitted bid `$b_t$` and update dual variable `$\lambda_t$` from Algorithm 1 based on `$\bar{F}_m$`, while the actual competing bids are generated according to `$F$`. */`
 4:      $T_m$ rounds of maximum competing bids $\{d_i : i = 2^{m-1}, ... 2^m - 1\}$ are revealed.
 5:      $N_{m+1} \leftarrow T_m + N_m$
 6:      Optimism parameter $\varepsilon_{m+1} \leftarrow \frac{\log(N_{m+1})}{\sqrt{N_{m+1}}}$.
 7:      Update the empirical CDF $\widehat{F}_m$ to be $\widehat{F}_{m+1}$ with the observed maximum competing bids $\{d_i : i = 1, ..., 2^m - 1\}$ according to Eqn. (5).
 8:      Obtain $\bar{F}_{m+1}$ defined as in Eqn. (6)
 9: **end for**
10: **return** The bid sequence $\{b_t\}_{t \in [T]}$.

---

**Algorithm performance.** We summarize the performance of Algorithm 2 as follows.

**Theorem 4.1.** *For any unknown competing bid distribution $F$ and unknown value distribution $H$, under the setting with full feedback, the expected regret of Algorithm 2 is bounded by* $\text{Regret}(\text{Algorithm 2} \mid (F, H)) = \widetilde{O}(\sqrt{T})$*, and the expected ROI violation is bounded by* $\Delta(\text{Algorithm 2} \mid (F, H)) = \widetilde{O}(\sqrt{T})$*.*

We conclude this section by providing a proof sketch for Theorem 4.1. All missing proofs with technical details are deferred to the full version [16].

*Proof of Theorem 4.1.* We first discuss the regret and ROI violation incurred with any fixed value sequence $V^T$. The analysis of regret and ROI violation of algorithm 2 consists of three main steps:

- **Step 1 – Accuracy of the optimistic CDF $\bar{F}$:** In this step, we proved that when $\varepsilon_m = \Theta\left(\frac{\log N_m}{\sqrt{N_m}}\right)$, the CDF $\bar{F}_m$ (defined in Eqn. (6)) serves as an optimistic estimate to the true underlying competing bid distribution ($\bar{F}_m \geq F$ with high probability), while still remains close, specifically we have $\sup_b |\bar{F}_m(b) - F(b)| \leq 2\varepsilon_m$ with high probability.

---

[10]As a shorthand, we write $x \wedge y := \min\{x, y\}$.

- **Step 2 – The performance approximation when implementing Algorithm 1 with estimated distributions** : In this step, we establish the performance guarantee (evaluating in an environment with competing bid distribution $F$) of the bids generated by feeding Algorithm 1 with a distribution $F^\dagger$ that satisfies $F^\dagger \geq F$ and $\sup_b |F_m^\dagger(b) - F(b)| \leq 2\varepsilon$,

$$\mathrm{Regret}(\mathsf{ALG1}(F^\dagger) \mid (F, V^T)) = O(\sqrt{T}) + 2\varepsilon T, \ \Delta(\mathsf{ALG1}(F^\dagger) \mid (F, V^T)) = \widetilde{O}(\sqrt{T}) + 2\varepsilon T .$$

  Here we denote by $\mathsf{ALG1}(F^\dagger)$ the bids generated by feeding Algorithm 1 with a distribution $F^\dagger$. In stage $m$, when we use the estimated optimistic $\bar{F}_m$ as input for Algorithm 1, both the regret and ROI violation will be amplified by $T \cdot 2\varepsilon_m$ compared to using the true CDF $F$ as input.
- **Step 3 – Aggregating the regret and ROI violation over $M$ stages** : With $\varepsilon_m$ selected in Step 1, the regret of Algorithm 2 is bounded by $\widetilde{O}(\sqrt{T})$. The ROI violation is bounded by $\widetilde{O}(\sqrt{T})$.

Thus, when a value sequence $V^T$ is i.i.d. drawn from $H^T$, the expected regret and ROI violation over the distribution $F$ and $H$ satisfies

$$\mathrm{Regret}(\text{Algorithm 2} \mid (F, H)) = \mathbb{E}_{V^T \sim H^T}\big[\mathrm{Regret}(\text{Algorithm 2} \mid (F, V^T))\big] = \widetilde{O}(\sqrt{T}) .$$

$$\Delta(\text{Algorithm 2} \mid (F, H)) = \mathbb{E}_{V^T \sim H^T}\big[\Delta(\text{Algorithm 2} \mid (F, V^T))\big] = \widetilde{O}(\sqrt{T}) . \qquad \square$$

## 5   No-Regret Autobidding Algorithm with Bandit Feedback

In this section, we consider a more challenging learning setting in which the bidder receives only bandit feedback at the end of each round – specifically, whether they won the item or not. This bandit feedback model is common in practice and has been extensively studied in the literature (see, e.g., [3, 15]). The main results in this section are an online autobidding algorithm (see Algorithm 3) with the provable performance (see Theorem 5.1) under such bandit feedback.

**Algorithm descriptions.**   At a high-level, our proposed Algorithm 3 operates as an explore-then-exploit type algorithm. The algorithm starts with an exploration stage that explores different bids to obtain sufficient knowledge about the underlying competing bid distribution. Then, the algorithm implements a "conservative" variant of Algorithm 1 with the estimated competing bid distribution over the remaining rounds.

In the exploration phase, to explore different bids, we discretize the bid range $[0, 1]$ into $K$ equal intervals with grid points $\{c_k\}_{k \in [K] \cup \{0\}}$, where $c_k = k/K$ for $k \in [K] \cup \{0\}$. We then bid each grid point $c_k$ consecutively for $M$ rounds, and then use the observed bidding outcomes to compute an empirical estimate $\widehat{F}$ for the underlying competing bid distribution $F$: for $\ell \in [1 : K - 1]$

$$\widehat{F}(c_\ell) = \max_{k \in [\ell] \cup \{0\}} \frac{1}{M} \cdot \sum_{t \in [k \cdot M + 1 : (k+1) \cdot M]} \mathbf{1}\{b_t \geq d_t\} , \tag{7}$$

and we let $\widehat{F}(c_K) = 1$. Note that in the bandit feedback setting, $d_t$ is unknown and only the indicator of winning or not, denoted as $\mathbf{1}\{b_t \geq d_t\}$, is revealed. We also extend the definition of $\widehat{F}$ to any bid $b \in [0, 1)$ as a step function: $\widehat{F}(b) = \widehat{F}(c_k)$ if $b \in [c_k, c_{k+1})$. After exploring all the grid bids, we can have sufficient observations to maintain a good estimate $\widehat{F}$ for the underlying the competing bid distribution. Similar to Algorithm 2, we also define the optimistic CDF $\bar{F}$ as follows:

$$\bar{F}(b) = \left(\widehat{F}(c_k) + \varepsilon\right) \wedge 1, \quad \text{if } b \in [c_k, c_{k+1}) , \tag{8}$$

where $\varepsilon$ again is a parameter carefully chosen to balance the exploration and exploitation tradeoff.

In Algorithm 2 with full feedback, after obtaining the optimistic CDF $\bar{F}$, we directly implement Algorithm 1 assuming the underlying competing bid distribution is $\bar{F}$. However, this approach may fail under our bandit feedback setting, as we can no longer guarantee a uniform error bound on $\bar{F}$ (i.e., $\sup_b |F(b) - \bar{F}(b)|$ might be large). Instead, in the exploitation phase of Algorithm 3, we implement Algorithm 1 for the remaining rounds in a "conservative" way. In particular, we define a conservative version of the optimistic CDF $\bar{F}^{\mathsf{c}}$ as follows:

$$\bar{F}^{\mathsf{c}}(b) = \bar{F}\left((b + 1/K) \wedge 1\right), \quad b \in [0, 1] . \tag{9}$$

We then feed Algorithm 1 with the input $T - K \cdot M$ as the time horizon, and $\bar{F}^{\mathsf{c}}$ as if the underlying competing bid distribution is $\bar{F}^{\mathsf{c}}$ to generate a sequence of bids $\{b_t^{\mathsf{c}}\}$ for the remaining rounds. Instead of directly using these bids, we submit the bids also in a conservative way. In particular, we bid $\{b_t\}$ where each bid $b_t \leftarrow (b_t^{\mathsf{c}} + 1/K) \wedge 1$.

---

**Algorithm 3** No-regret bidding algorithm with bandit feedback

---

**Input:** Time horizon $T$, the value sequence $V^T$ (revealed sequentially).
**Initialize:** Initialize dual variable $\lambda_1 = 1, K, M$.

    /* Exploration phase */
1: Discretize $[0, 1]$ to obtain grid points $\{c_0, c_1, c_2, \ldots, c_K\}$.
2: **for** $k \leftarrow 0$ to $K - 1$ **do**
3:     Bid $c_k$ consecutively for $M$ time rounds.
4:     Observe the allocation outcomes $\mathbf{1}\{b_t \geq d_t\}_{t \in [k \cdot M + 1 : (k+1) \cdot M]}$.
5: **end for**
6: Obtain the empirical CDF $\widehat{F}$, the optimistic CDF $\bar{F}$ and the conservative CDF $\bar{F}^{\mathsf{c}}$ defined as in Eqn. (7), (8) and (9), respectively.
    /* Exploitation phase */
7: Feed the Algorithm 1 with inputs: $T \leftarrow T - K \cdot M, F \leftarrow \bar{F}^{\mathsf{c}}$.
8: Let $\{b_t^{\mathsf{c}}\}$ be the bids generated from Algorithm 1 with these input specifications.
9: Submit the bids $\{b_t\}$ over remaining rounds where each bid $b_t \leftarrow (b_t^{\mathsf{c}} + 1/K) \wedge 1$.
10: **return** The bid sequence $\{b_t\}_{t \in [T]}$

---

**Algorithm performance.** The performance of Algorithm 3 is summarized as follows:

**Theorem 5.1.** *For any unknown competing bid distribution $F$ and unknown value distribution $H$, under the bandit feedback, the expected regret of Algorithm 3 with $M = \Theta(\sqrt{T}), K = \Theta(T^{1/4})$ is bounded by* $\mathrm{Regret}(\text{Algorithm } 3 \mid (F, H)) = \widetilde{O}(T^{3/4})$, *and the expected ROI violation is bounded by* $\Delta(\text{Algorithm } 3 \mid (F, H)) = \widetilde{O}(T^{3/4})$.

The proof of Theorem 5.1 and all relevant lemmas (with proofs) are deferred to the full version[16].

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
