# OpenReview forum: "No-Regret Online Autobidding Algorithms in First-price Auctions"
_NeurIPS.cc/2025/Conference — NeurIPS 2025 poster_

### Official Review · Reviewer_CqEZ · 2025-06-25

**Clarity:** 2
**Significance:** 3
**Originality:** 3
**Rating:** 4
**Confidence:** 3

**Summary:**

The paper presents online autobidding algorithms for repeated first-price auctions with ROI constraints. These algorithms compete against the optimal randomized strategy in hindsight. More specifically, the authors consider the problem where a single bidder participates in single-item first-price auctions repeated over $T$ rounds. At each time step, the learner observes the value $v_t$​ and then submits a bid $b_t$ based on this value and the history. The bidder wins the item if her submitted bid is not less than the highest competing bid  $d_t$ in that round. If she wins, she pays her submitted bid; otherwise, she pays nothing. The goal is to design an online bidding algorithm that maximizes the total realized utility, subject to a Return-On-Investment (ROI) constraint, meaning the total utility should be greater than the total payment. The authors first focus on the offline optimal strategy, which may be randomized, and show—using a convexification argument on the allocation-payment curve—that this randomized problem can be reduced to a deterministic one by replacing the original allocation function with its concave envelope. Based on this, they develop Algorithm 1 for the case where the distribution of the competition is known, and prove that it achieves a regret of order $O(T^{1/2})$. When the distribution of the competition is unknown, they consider two settings: full feedback, where the bidder observes the competing bid $d_t$ at the end of each round, and bandit feedback, where the bidder only observes a binary win/loss signal (0 or 1). For the full feedback setting, they develop an algorithm based on mirror descent that achieves a regret of $O(T^{1/2})$, and for the bandit setting, they propose an explore-then-commit strategy that achieves a regret of $O(T^{3/4})$, leaving a gap with the known lower bound for this problem $\Omega(T^{2/3})$.

**Questions:**

- Could the authors provide intuitions on how the problem or solution would change if a budget constraint were introduced?
- Are there any ideas or insights on how the regret bound in the bandit feedback setting might be improved ?
- Can the authors discuss how the setting and results might extend to the more realistic case where multiple bidders are learning simultaneously, as is common in online advertising ?

Minor comment: There appears to be a formatting typo at the end of page 9 in the expressions for $T^{1/4}$ and $T^{3/4}$, where the closing brace is missing or misplaced.

**Ethical Concerns:**

["NO or VERY MINOR ethics concerns only"]

**Final Justification:**

Given the discussion with the authors, I am confident in my first opinion about the paper.  I therefore maintain my rating for this paper.

**Limitations:**

yes

**Paper Formatting Concerns:**

No paper formatting concerns.

**Quality:**

3

**Strengths And Weaknesses:**

Strengths:
- The paper is well motivated and addresses an important problem in online advertising.
- From a technical standpoint, the author use a key convexification argument to reduce the randomized constrained optimization problem of the optimal strategy (OPT) to an equivalent deterministic optimization problem.
- When the distribution of the bids of the competition is known, they propose an algorithm based on this convexification argument that achieves a regret of $O(T^{1/2})$.
- When the distribution of the bids of the competition is unknown, and the bidder has a full feedback (the competing bid) at the end of the round, they propose an algorithm based on mirror descent that achieves a regret of order $O(T^{1/2})$.
- In the more challenging bandit feedback setting—where only a binary win/loss signal is observed, they propose an algorithm based on explore-then-commit strategy that achieves a regret of order $O(T^{3/4})$,

Weaknesses:
- The paper lacks numerical simulations to empirically demonstrate the performance of the proposed algorithms.
- Although the authors acknowledge the remaining gap in the regret bound for the bandit setting, the paper does not propose potential directions to reduce this gap.
- The explanation of why the convexification argument preserves the optimal solution could be elaborated; in particular, the transition from optimizing under the original distribution $F$ to its concave enveloppe $F_{conv}$ needs more intuitions.
- While the paper focuses on ROI constraints, which are important, it does not address how budget constraints—commonly present in real-world bidding environments—could be integrated into the proposed framework.

---

> ### Author Rebuttal · Authors · 2025-07-31
>
> Thank you for your detailed and insightful feedback.
>
> > Weakness 1: The paper lacks numerical simulations to empirically demonstrate the performance of the proposed algorithms.
>
> We agree that numerical experiments would be generally helpful. Nonetheless, we did not perform them for two main reasons: (1) The paper is already fairly packed with the main theoretical results (with almost all proofs deferred to appendices), and we figured it would serve the audience better to focus on the theory part and convey the main ideas and messages properly. (2) To better quantify the practical feasibility of our results, ideally, we'd run experiments on real-life data. Such data, unfortunately, is unavailable due to the nature of the online advertising industry (regulation, privacy, business considerations, etc.).
>
> > Weakness 2: Although the authors acknowledge the remaining gap in the regret bound for the bandit setting, the paper does not propose potential directions to reduce this gap.
>
> > Question 2: Are there any ideas or insights on how the regret bound in the bandit feedback setting might be improved?
>
> As you point out, we were unable to determine the optimal regret bound for the bandit-feedback setting, but not for lack of trying -- for what it's worth, Aggarwal et al. [2] couldn't close the gap either despite the fact that they investigated a weaker benchmark than ours. As for directions for improvement: if we had to make a guess, we'd say the lower bound by Aggarwal et al. [2] is more likely to be tight. On the other hand, the exploration-exploitation approach doesn't seem to have the potential to give a better bound than $O(T^{3/4})$ (e.g., our algorithm has an exploration phase of length $T^{3/4}$, which already generates regret of order $T^{3/4}$). So, to improve the upper bound, one would likely need to take a fundamentally different approach.
>
> > Weakness 3: The explanation of why the convexification argument preserves the optimal solution could be elaborated; in particular, the transition from optimizing under the original distribution $F$ to its concave envelope $F_{conv}$ needs more intuition.
>
> The proof itself is not straightforward, and we had to place (most of) it in an appendix. Here is the intuition behind the proof: to understand the structure of optimal strategies, it is more convenient to work with the allocation-payment curve $G$, rather than the distribution $F$ of competing bids itself. The allocation-payment curve $G$ can be viewed as a parametrized curve by the bid $b$: when the algorithm bids $b$, the probability of winning (i.e., the allocation) is $F(b)$, and the expected payment is $F(b) \cdot b$, so $(F(b), F(b) \cdot b)$ is a point on the curve. Letting $b$ vary across all possible bids in $[0, 1]$, we get a curve that describes the relation between the allocation and the amount of payment required to achieve the allocation, when the algorithm bids deterministically. Now to incorporate randomized bidding, observe that randomizing between two bids corresponds to taking a convex combination of two points on the allocation-payment curve. As a result, the best tradeoff between allocation and payment under randomized bidding is given by the concave envelope $G_{conv}$ of the allocation-payment curve $G$. Now observe that if there is another distribution $F_{conv}$ whose corresponding allocation-payment curve happens to be $G_{conv}$, then deterministic bidding strategies on $F_{conv}$ are "as good as" randomized bidding strategies on $F$. We show that this is in fact always the case: $F_{conv}$ always exists and we can construct it from $F$. This finishes the proof. We will add a discussion in the paper if space allows.
>
> > Weakness 4: While the paper focuses on ROI constraints, which are important, it does not address how budget constraints—commonly present in real-world bidding environments—could be integrated into the proposed framework.
>
> > Question 1: Could the authors provide intuitions on how the problem or solution would change if a budget constraint were introduced?
>
> As mentioned in the paper, our results generalize naturally to settings with ROI and / or budget constraints. We did not discuss the generalization in detail, since the mirror-descent-based primitive is known to work naturally for both budget and ROI constraints in similar settings (see, e.g., the work by Feng et al. [1]), and to handle budget constraints instead of / in addition to ROI constraints in our setting, one would simply combine the budget-constrained version of the primitive with all the additional work done in our paper, including all steps discussed in our technical overview. When we add the budget constraint, the corresponding changes we need to make is similar to what Feng et al. [1] did in Section 5.1 of their paper, which roughly involves (1) introducing a new Lagrangian multiplier $\mu$ corresponding to the budget constraint, (2) updating $\mu$ in a way similar to how $\lambda$ is updated in our paper, and (3) terminating the algorithm when the remaining budget is too small. The way bids are computed needs to be adapted accordingly. More specifically, Eq. (2) should be modified into the following:
> $\max \mathbb{E} \left[\sum_{t\in[T]}  v_t \cdot F(b_t) + \min_{\lambda \ge 0} \lambda \cdot \sum_{t\in[T]} (v_t\cdot F(b_t) - b_t \cdot F(b_t)) + \min_{\mu \ge 0} \mu \cdot (B - \sum_{t\in[T]} b_t \cdot F(b_t))\right],$
> where $B$ is the total budget; bids should be computed in the following way:
> $\widetilde b_t \gets argmax_{b\in[0, 1]} \left[(1+\lambda_t)\cdot v_t\cdot F_{conv}(b)- (\lambda_t + \mu_t) \cdot b \cdot F_{conv}(b)\right].$
>
> Regarding the analysis of the regret bound, in Feng et al.'s paper, they point out that the combination of budget and ROI constraints is intuitively straightforward, since the analyses of the algorithms with the two types of constraints respectively are linear in nature. This means one can handle these two constraints separately, and taking the linear combination, the regret under both constraints is still $O(\sqrt{T})$. Overall, the adaptation would be somewhat repetitive technically (with respect to both our analysis of the case with ROI constraints, and Feng et al.'s work [1]), which we wanted to avoid because of limited space; rather, we chose to focus on the additional challenges posed by non-truthful auction mechanisms. We will add a more detailed discussion in the paper.
>
> > Question 3: Can the authors discuss how the setting and results might extend to the more realistic case where multiple bidders are learning simultaneously, as is common in online advertising?
>
> Put in our language, this essentially means the distribution of competing bids changes over time in a certain way (depending on the learning algorithm used by other bidders). To handle such cases, one straightforward idea would be to design no-regret algorithms in adversarial environments.  Unfortunately, this is impossible for the goal we are trying to achieve (i.e., competing against the optimal bidding strategy, rather than the optimal fixed bid). Alternatively, one might try to relax the stochastic model a little and consider, e.g., a model where the distribution of competing bids shifts *slowly* over time. In such a model, one could make easy modifications to our algorithms to achieve nontrivial guarantees, e.g., by pretending the competing bid distribution is stationary, and restarting the algorithm from scratch every once in a while when the distribution shift becomes significant enough.
>
> [1] Zhe Feng, Swati Padmanabhan, and Di Wang. Online Bidding Algorithms for Return-on-Spend Constrained Advertisers.
>
> [2] Gagan Aggarwal, Giannis Fikioris, and Mingfei Zhao. No-regret algorithms in non-truthful auctions with budget and ROI constraints.

---

> > ### Comment · Reviewer_CqEZ · 2025-08-02
> > **Thank you**
> >
> > Thank you for the detailed and thoughtful rebuttal. I appreciate the clarifications and insights you’ve provided. I will keep my current score for now and look forward to engaging in the discussion with the other reviewers.

---

### Official Review · Reviewer_Xvxb · 2025-06-30

**Clarity:** 3
**Significance:** 3
**Originality:** 4
**Rating:** 4
**Confidence:** 4

**Summary:**

This paper studies the autobidding problem under the setting of ROI-constrained value maximizers in first-price auctions. It considers two distinct feedback settings: (1) the full-feedback setting, where the maximum competing bid is observed, and (2) the bandit-feedback setting, where the bidder only observes whether they win each auction. For each setting, the paper develops corresponding bidding algorithms with regret guarantees.

**Questions:**

1. Do the value vectors $v_t$, for $t \in [T]$, need to be i.i.d. in your setting?
2. Should the input to Algorithm 2 include the maximum competing bids revealed sequentially?
3. How does definition (6) depend on $c_l$? $d_t$ should not be known in the bandit feedback setting.

**Ethical Concerns:**

["NO or VERY MINOR ethics concerns only"]

**Final Justification:**

The authors have partially addressed my concerns, but the practical feasibility and effectiveness may need to be further elaborated. I will maintain my current positive score and remain slightly inclined toward acceptance.

**Limitations:**

See  Weaknesses

**Paper Formatting Concerns:**

No.

**Quality:**

3

**Strengths And Weaknesses:**

Strength:
1. The paper addresses an important problem in online advertising and provides useful regret bounds in a novel setting that avoids some of the assumptions made in prior work.
2. The paper is well-written and well-organized.
           Weakness:
1. The paper considers only ROI constraints and does not incorporate budget constraints, which are important in practical settings.
2. The paper lacks numerical results and real-world applications, leaving open questions about the practical feasibility and effectiveness of the proposed approach.

---

> ### Author Rebuttal · Authors · 2025-07-31
>
> Thank you for your thoughtful and helpful feedback.
>
> > Weakness 1: The paper considers only ROI constraints and does not incorporate budget constraints, which are important in practical settings.
>
> As mentioned in the paper, our results generalize naturally to settings with ROI and / or budget constraints. We did not discuss the generalization in detail, since the mirror-descent-based primitive is known to work naturally for both budget and ROI constraints in similar settings (see, e.g., the work by Feng et al. [1]), and to handle budget constraints instead of / in addition to ROI constraints in our setting, one would simply combine the budget-constrained version of the primitive with all the additional work done in our paper, including all steps discussed in our technical overview. When we add the budget constraint, the corresponding changes we need to make is similar to what Feng et al. [1] did in Section 5.1 of their paper, which roughly involves (1) introducing a new Lagrangian multiplier $\mu$ corresponding to the budget constraint, (2) updating $\mu$ in a way similar to how $\lambda$ is updated in our paper, and (3) terminating the algorithm when the remaining budget is too small. The way bids are computed needs to be adapted accordingly. More specifically, Eq. (2) should be modified into the following:
> $\max \mathbb{E} \left[\sum_{t\in[T]}  v_t \cdot F(b_t) + \min_{\lambda \ge 0} \lambda \cdot \sum_{t\in[T]} (v_t\cdot F(b_t) - b_t \cdot F(b_t)) + \min_{\mu \ge 0} \mu \cdot (B - \sum_{t\in[T]} b_t \cdot F(b_t))\right],$
> where $B$ is the total budget; bids should be computed in the following way:
> $\widetilde b_t \gets argmax_{b\in[0, 1]} \left[(1+\lambda_t)\cdot v_t\cdot F_{conv}(b)- (\lambda_t + \mu_t) \cdot b \cdot F_{conv}(b)\right].$
>
> Regarding the analysis of the regret bound, in Feng et al.'s paper, they point out that the combination of budget and ROI constraints is intuitively straightforward, since the analyses of the algorithms with the two types of constraints respectively are linear in nature. This means one can handle these two constraints separately, and taking the linear combination, the regret under both constraints is still $O(\sqrt{T})$. Overall, the adaptation would be somewhat repetitive technically (with respect to both our analysis of the case with ROI constraints, and Feng et al.'s work [1]), which we wanted to avoid because of limited space; rather, we chose to focus on the additional challenges posed by non-truthful auction mechanisms. We will add a more detailed discussion in the paper.
>
> > Weakness 2: The paper lacks numerical results and real-world applications, leaving open questions about the practical feasibility and effectiveness of the proposed approach.
>
> We agree that experiments would be generally helpful. Nonetheless, we did not perform them for two main reasons: (1) The paper is already fairly packed with the main theoretical results (with almost all proofs deferred to appendices), and we figured it would serve the audience better to focus on the theory part and convey the main ideas and messages properly. (2) To better quantify the practical feasibility of our results, ideally, we'd run experiments on real-life data. Such data, unfortunately, is unavailable due to the nature of the online advertising industry (regulation, privacy, business considerations, etc.).
>
> As for real-world applications: our results (as well as the most related prior work) can be naturally applied in online advertising -- in fact, this is the very motivation for our study. Major players such as Google and Meta are known to run algorithms of this kind (i.e., autobidders) on their advertising platforms on behalf of advertisers. We cannot provide URLs in the response, but relevant information can be found on the respective websites of these platforms.
>
> > Question 1: Do the value vectors $v_t$, for $t \in [T]$ need to be i.i.d. in your setting?
>
> Yes, we assume they are i.i.d., as Feng et al. [1] and Aggarwal et al. [2] do in their work. This is in some sense necessary, since one wouldn't be able to get sublinear regret if these values are adversarial.
>
> > Question 2: Should the input to Algorithm 2 include the maximum competing bids revealed sequentially?
>
> Yes, you are right. We apologize for the confusion. The maximum competing bids should be revealed sequentially, and the sequence of realized competing bids are used in Eq. (4) to update the empirical distribution iteratively. We will make this more explicit in the paper.
>
> > Question 3: How does definition (6) depend on $c_l$? $d_t$ should not be known in the bandit feedback setting.
>
> In Eq. (6) (assuming this is what you referred to), $F(c_\ell)$ depends on $c_\ell$ through the maximization over $k \in [\ell] \cup \\{0\\}$. Moreover, $F(c_\ell)$ only depends on the indicator of winning or not ($\mathbb{1}[b_t \ge d_t]$). While $d_t$ is unknown to us, this indicator is precisely the one-bit response we observe in the bandit feedback setting. We will clarify in the paper.
>
> [1] Zhe Feng, Swati Padmanabhan, and Di Wang. Online Bidding Algorithms for Return-on-Spend Constrained Advertisers.
>
> [2] Gagan Aggarwal, Giannis Fikioris, and Mingfei Zhao. No-regret algorithms in non-truthful auctions with budget and ROI constraints.

---

### Official Review · Reviewer_xKcP · 2025-06-30

**Clarity:** 3
**Significance:** 2
**Originality:** 2
**Rating:** 4
**Confidence:** 3

**Summary:**

This paper focuses on the design and analysis of autobidding algorithms for online first-price auctions. More specifically, the focus is on the problem of bidding repeatedly under a certain Return Over Investment constraint. This is motivated by the fact that these kinds of auctions and of constraints are common practice in online advertising auctions. The main challenge in this problem comes from the fact that the opposing bid distribution $F$, and the distribution of the agents' value $H$ are unknown and need to be learned online. This paper considers two types of feedback after each round: full information and bandit feedback. Before stating their algorithm, the paper provides a reduction of their problem against an arbitrary distribution $F$ to a problem against a concavified version $F$, which simplifies the problem and ensures known techniques can be better adapted. They then provide an algorithm, regret, and ROI constraint violation guarantees for this algorithm. They show their algorithm can guarantee a regret upper bounded by $\tilde{\mathcal{O}}\left ( \sqrt{T} \right )$ under full-information feedback and $\tilde{\mathcal{O}}\left ( T^{3/4} \right )$ under bandit feedback. This result improves on the literature, not in terms of regret rates but on the class of comparator strategies against which the regret is computed (General vs Lipschitz).

**Questions:**

1. I am wondering about the link with auctions with replenishing budget. It does seem that if we consider the setting where an algorithm for bidding is constrained by its budget but receives additional budget (say $\rho v^t$ when the auction is won), an algorithm should be similar to the one in your setting? A setting of this kind is fully described in [1].

2. I understood that once the transformation (concavification of $F$) is done, the best strategy for each valuation $v$ is a deterministic bid. Therefore, we could define a function which maps the value observed to the best bid corresponding $b(v)$. I am wondering whether this function would be Lipschitz (or maybe only right/left Lipschitz). Do you have any counterexamples? I am asking to understand better the scale of the contribution. If the best in hindsight function $b(v)$ in the concavified version of the problem is Lipschitz, then one could use the result from previous work, right?
(I understand that making the link between the concavified and original problem and showing how uncertainty in one is related to uncertainty in the other would remain an issue.)

3. You did not provide in the paper any result on lower bounds. It seems from the theoretical results alone that it is unclear whether the tight regret bound should be $T^{2/3}$ or $T^{3/4}$. Do you have any insight? Did you run experiments for which a regret of $T^{2/3}$ could be attained?

4. Is it clear that the optimal bound in the full-information feedback is $\mathcal{O} ( \sqrt{T} )$? Is there an existing lower bound in the literature, or could such a result be derived?


[1] Berriaud, Damien, et al. "To Spend or to Gain: Online Learning in Repeated Karma Auctions." arXiv preprint arXiv:2403.04057 (2024).

**Ethical Concerns:**

["NO or VERY MINOR ethics concerns only"]

**Final Justification:**

Given the discussion with the authors, I am confident in my first opinion about the paper. Furthermore, I believe the weakness we discussed the most can be easily addressed. I therefore maintain my rating for this paper.

**Limitations:**

yes

**Paper Formatting Concerns:**

no concerns

**Quality:**

3

**Strengths And Weaknesses:**

This paper presents a well-motivated and technically sound approach to autobidding in first-price auctions under ROI constraints, offering improved regret guarantees in both full-information and bandit settings, though it would benefit from lower bounds and empirical validations.

# Strenght :

- The paper is well written and motivated. The problem studied is interesting and the tools needed to obtain the result are non-trivial.

- The paper proves new results that do not immediately follow from known ones.

- The paper is technically sound and includes a helpful overview of the technical challenges, both in the introduction and in the proof sketches.

# Weaknesses :

- The paper doesn't provide matching lower bounds for the result stated. This makes it difficult to assess whether the rates obtained are optimal or if there is still room for improvement.

- The paper lacks experiments. It would specifically be nice to have some in order to compare their result with the work of Aggarwal et al. [3], which is referred to in the paper. Their result strengthens this result in a non-straightforward manner (by extending the family of functions to which the fixed best strategy used to compute the regret belongs), as a result, it is difficult to gauge the difference made by this change. Experiments would provide a nice way to give some intuition.

- The paper states that their result easily generalizes to other auction mechanisms (line 49, and footnote 4, page 4) as part of their result. It would be preferable to avoid emphasizing this claim without proof. I would be more comfortable if that was mentioned in the introduction and or conclusion (not in result / setting sections). Otherwise, you could provide (eventually in the appendix) a note describing how this would generalize to a few specific common auctions.

---

> ### Author Rebuttal · Authors · 2025-07-31
>
> Thank you for your detailed and thoughtful feedback.
>
> > Weakness 1: The paper doesn't provide matching lower bounds for the result stated. This makes it difficult to assess whether the rates obtained are optimal or if there is still room for improvement.
>
> > Question 3: You did not provide in the paper any result on lower bounds. It seems from the theoretical results alone that it is unclear whether the tight regret bound should be $T^{2/3}$ or $T^{3/4}$. Do you have any insight? Did you run experiments for which a regret of  $T^{2/3}$ could be attained?
>
> > Question 4: Is it clear that the optimal bound in the full-information feedback is $\Omega(\sqrt{T})$? Is there an existing lower bound in the literature, or could such a result be derived?
>
> Let us respond to the 3 closely connected questions together. In the full feedback setting, there is a folklore lower bound of $\Omega(\sqrt{T})$, so our upper bound of $\tilde{O}(\sqrt{T})$ is almost tight. The folklore lower bound is essentially inherited from the most fundamental setting of stochastic bandits, even with only two discrete actions. The idea is that to distinguish two actions whose expected rewards are within $\Delta$ of each other, one roughly needs $\Omega(\Delta^{-2})$ rounds to be confident enough. So after $T$ rounds, intuitively, one would be able to find an action that is optimal up to an additive error of $\Theta(1 / \sqrt{T})$. This translates to a cumulative error of $\Theta(T \cdot 1 / \sqrt{T}) = \Theta(\sqrt{T})$. To instantiate this bound in our setting, one could, for example, consider the following setup: the value distribution is supported on $\\{1/3, 2/3\\}$, whose mean is either $1/2 - \varepsilon$ or $1/2$; the competing bid distribution is a point mass at $1/2$, i.e., the competing bid is always $1/2$. The optimal bidding strategy, depending on the mean of the value distribution, is either (1) always bidding $1/2$ (when the mean is $1/2$), or (2) bidding $1/2$ deterministically when the value is $2/3$, and bidding $1/2$ with probability $1 - \Theta(\varepsilon)$ (and bidding $0$ otherwise) when the value is $1/3$. But to figure out which one is better, the algorithm must first estimate the mean of the value distribution accurately enough, which takes $\Omega(\varepsilon^{-2})$ rounds. By arguments similar to the one sketched above, one can establish an $\Omega(\sqrt{T})$ lower bound. We will add a remark in the paper.
>
> In the bandit feedback setting, Aggarwal et al.'s work (reference [3] in the paper) establishes a stronger lower bound of $\Omega(T^{2/3})$, which in particular means the bandit feedback setting is strictly harder. Unfortunately, as you point out, we don't have much evidence supporting either $T^{2/3}$ or $T^{3/4}$ as the "right" bound. If we had to take a guess, we would tend to pick $T^{2/3}$, since the lower bound construction by Aggarwal et al. appears quite difficult to improve. If that is the case, then progress on the upper bound side would probably require a non-exploration-exploitation approach. In particular, our algorithm by design must suffer a regret of $\Omega(T^{3/4})$, because of the exploration phase of length $\Omega(T^{3/4})$.
>
> As for numerical experiments: we did not try them, because (1) our algorithm by design cannot perform better than $\Omega(T^{3/4})$, as discussed above, and (2) even if, hypothetically, the "right" bound was $\Theta(T^{3/4})$, it would be difficult to spot a hard instance that forces this bound via random guess. Presumably, "common" instances would be easy, i.e., we would almost always end up with "easy" instances if we generate them randomly in a natural way (the hard instance forcing $\Omega(T^{2/3})$ by Aggarwal et al. is fairly delicate). Experimenting on such instances likely wouldn't provide much insight.
>
> > Weakness 2: The paper lacks experiments. It would specifically be nice to have some in order to compare their result with the work of Aggarwal et al. [3] ...
>
> We agree that experiments would be generally helpful. Nonetheless, we did not perform them for two main reasons: (1) The paper is already fairly packed with the main theoretical results (with almost all proofs deferred to appendices), and we figured it would serve the audience better to focus on the theory part and convey the main ideas and messages properly. (2) As you point out, the improvement over Aggarwal et al.'s result is non-straightforward, particularly in the sense that on some instances, the improvement is extremely large, while on others, it is much less significant. So if we were to run experiments, we'd ideally run them on real-life data to better quantify the practical improvement. Such data, unfortunately, is unavailable due to the nature of the online advertising industry (regulation, privacy, business considerations, etc.).
>
> > Weakness 3: The paper states that their result easily generalizes to other auction mechanisms (line 49, and footnote 4, page 4) as part of their result. It would be preferable to avoid emphasizing this claim without proof ...
>
> Thanks for the suggestion. The generalization involves mostly the characterization in Section 3.1. Recall that the current characterization turns the input competing bid distribution into a "concavified" version, by concavifying the allocation-payment curve corresponding to the competing bid distribution and mapping it back. If one were to use another auction mechanism, the mapping between the competing bid distribution and the allocation-payment curve would change depending on the allocation and payment rules of the auction mechanism, but other than that, one would be able to perform essentially the same operation on the competing bid distribution. After this step, one only needs to solve the Lagrangified value maximization problem under the new auction mechanism (corresponding to the current Eq. (2) and (3)). Again, the specific form of the objective here depends on the allocation and payment rules of the mechanism, but other than that, everything in Algorithm 1 remains exactly the same. No further changes are needed in Algorithms 2 and 3, which simply call the modified Algorithm 1 as a subroutine. We will add a discussion of this (possibly as an appendix).
>
> > Question 1: I am wondering about the link with auctions with replenishing budgets. It does seem that if we consider the setting where an algorithm for bidding is constrained by its budget but receives additional budget (say $\rho v^t$ when the auction is won), an algorithm should be similar to the one in your setting? A setting of this kind is fully described in [1].
>
> This is an interesting insight. The way we view this connection is it provides a different perspective on the problem, though it's not immediately clear how it might help technically. The setting in [1] in particular seems to differ from ours in a number of ways (equally redistributed "karma" among agents, step-wise budget feasibility constraints, etc.), though of course the conceptual connection (non-truthfulness of second-price auctions, etc.) is interesting. It would also be interesting to see if one can unify these models into a single framework. We will discuss this in the paper.
>
> > Question 2: I understood that once the transformation (concavification of $F$) is done, the best strategy for each valuation $v$ is a deterministic bid. Therefore, we could define a function which maps the value observed to the best bid corresponding $b(v)$. I am wondering whether this function would be Lipschitz (or maybe only right/left Lipschitz). Do you have any counterexamples? I am asking to understand better the scale of the contribution. If the best in hindsight function $b(v)$ in the concavified version of the problem is Lipschitz, then one could use the result from previous work, right? ...
>
> This is indeed a natural idea. Unfortunately, the mapping you mention (value to optimal bid after concavification) is not always Lipschitz: consider a value distribution supported on $\\{1/2 - \varepsilon, 1/2 + \varepsilon \\}$ for some small $\varepsilon > 0$ (probabilities to be fixed later), and a competing bid distribution supported on $\\{0, 1\\}$, say with equal probabilities. One can tune the probabilities on $1/2 - \varepsilon$ and $1/2 + \varepsilon$ respectively, such that the mapping $1/2 - \varepsilon \mapsto 0, 1/2 + \varepsilon \mapsto 1$ is the unique optimal strategy (one only needs to make sure the ROI constraint is binding under this strategy). This strategy can be highly "not continuous" when $\varepsilon$ is small. Letting $\varepsilon \to 0$ then rules out any finite Lipschitz constant. In other words, at least one cannot apply techniques from Lipschitz bandits in a straightforward way.
>
> Independent of Lipschitz continuity, we'd also remark that the choice of the innermost primitive is technically insignificant. Most technical work in the paper lies in reducing and transforming the problem to a form that can be handled using existing machinery -- in our paper, we chose to use the mirror-descent-based primitive. The adaptation we made for the primitive to work in the exact way we need it to is relatively minor compared to the rest of the reduction. Suppose, hypothetically, that the mapping discussed above was in fact Lipschitz, then one would be free to choose either the mirror-descent-based primitive or the Lipschitz bandits primitive as the innermost block, which, in our opinion, wouldn't affect the nature of the algorithm. However, even in this hypothetical world, one still wouldn't be able to run the Lipschitz bandits primitive in an end-to-end fashion (i.e., as a standalone algorithm), because without knowing the competing bid distribution, one wouldn't be able to interpret "deterministic" bids generated by the primitive, or even compute effective rewards.

---

> > ### Comment · Reviewer_xKcP · 2025-08-03
> >
> > Thank you for your response.
> >
> > ### Lower bound $\sqrt{T}$.
> > I agree that obtaining a lower bound in $\Omega \left ( \sqrt{T} \right )$ for this problem seems like it should be straightforward or easily obtained by reduction to a similar problem (which often exhibits these kinds of lower bounds in full-information). I was mostly pointing out that this isn't mentioned in the paper and should. It should be accompanied by a quick proof sketch or ideally by a reduction to a similar problem for which you could reference the lower bound.
> >
> > ### Auction with replenishing budget.
> > I mentioned this other work as it seemed similar to me. I want to highlight the fact that discussing this other work and its link to this one is not a required addition to the paper. I will let you be the judge of whether or not it is relevant.

---

> > > ### Author Response · Authors · 2025-08-06
> > >
> > > **Re $\sqrt{T}$ lower bound**: thank you for pointing this out.  The lower bound certainly deserves a more formal discussion, and we will add one in the paper.
> > >
> > > **Re replenishing budget**: thanks again for bringing this line of work to our attention.  We believe there is a nontrivial and potentially interesting connection, and discussing it would only make the paper more complete.  We will add a brief note in the related work part.

---

### Official Review · Reviewer_jSPN · 2025-07-04

**Clarity:** 3
**Significance:** 2
**Originality:** 2
**Rating:** 4
**Confidence:** 4

**Summary:**

The paper addresses the design of online bidding algorithms for repeated first-price auctions with ROI constraints. It benchmarks performance against the optimal randomized strategy in hindsight, focusing on two feedback settings: full feedback (where the highest competing bid is observed) and bandit feedback (where only win/loss outcomes are known). The goal is to minimize regret and ensure the ROI constraint is met.

**Questions:**

NA

**Ethical Concerns:**

["NO or VERY MINOR ethics concerns only"]

**Final Justification:**

The author clearified how to deal with budget constraint, which addressed part of my concerns. I am positive on this paper now, although still think it is a boardline paper.

**Paper Formatting Concerns:**

No formatting issue.

**Quality:**

2

**Strengths And Weaknesses:**

Strengths:
1. The result of near-optimal algorithm for the repeated first-price auctions with ROI constraints is important.
2. Concrete theoretical analysis of the regret bounds for the proposed algorithm.

Weaknesses:
1. The algorithm design is kind of straightforward. Basically, it is built on an optimal algorithm requiring known distribution, and a subtle process dealing with estimating the unknown distribution.
2. The budget constraint is not considered. Although it is claimed that the result can be generalized to the setting with budget constraints, it seems not to be trivial.

---

> ### Author Rebuttal · Authors · 2025-07-31
>
> Thank you for your insightful and constructive feedback.
>
> > Weakness 1: The algorithm design is kind of straightforward. Basically, it is built on an optimal algorithm requiring known distribution, and a subtle process dealing with estimating the unknown distribution.
>
> As discussed in our technical overview, there are several concrete technical challenges: characterizing ostensibly unstructured optimal strategies, reconciling the mirror-descent-based primitive and our characterization, dealing with unknown distributions of competing bids, etc. While an individual step in our roadmap might appear natural in hindsight, it was certainly not straightforward that this was the right path towards an almost optimal algorithm before the fact. Also, as we compare with the most relevant prior work in our paper, Feng et al. [1] study the problem with truthful auction mechanisms, where the distribution of competing bids is irrelevant, and Aggarwal et al. [2] design the algorithm against a weaker benchmark -- the best Lipschitz bidding function that maps values to bids. Both prior results, which already bear a remarkable amount of technicality, solve only special / weaker cases of the problem under restrictive conditions. Given that, we would argue that even if we disregard all technical details, the fact that we manage to handle non-truthful actions and compete against unconditionally optimal benchmarks is evidence that there is something nontrivial going on. Further details can be found in the technique overview part of the paper.
>
> > Weakness 2: The budget constraint is not considered. Although it is claimed that the result can be generalized to the setting with budget constraints, it seems not to be trivial.
>
> We are sorry for being overly brief. The reason that we (implicitly) stated that the generalization is "easy" is that the mirror-descent-based primitive is known to work naturally for both budget and ROI constraints in similar settings (see, e.g., the work by Feng et al. [1]), and to handle budget constraints instead of / in addition to ROI constraints in our setting, one would simply combine the budget-constrained version of the primitive with all the additional work done in our paper, including all steps discussed in our technical overview. When we add the budget constraint, the corresponding changes we need to make is similar to what Feng et al. [1] did in Section 5.1 of their paper, which roughly involves (1) introducing a new Lagrangian multiplier $\mu$ corresponding to the budget constraint, (2) updating $\mu$ in a way similar to how $\lambda$ is updated in our paper, and (3) terminating the algorithm when the remaining budget is too small. The way bids are computed needs to be adapted accordingly. More specifically, Eq. (2) should be modified into the following:
> $\max \mathbb{E} \left[\sum_{t\in[T]}  v_t \cdot F(b_t) + \min_{\lambda \ge 0} \lambda \cdot \sum_{t\in[T]} (v_t\cdot F(b_t) - b_t \cdot F(b_t)) + \min_{\mu \ge 0} \mu \cdot (B - \sum_{t\in[T]} b_t \cdot F(b_t))\right],$
> where $B$ is the total budget; bids should be computed in the following way:
> $\widetilde b_t \gets argmax_{b\in[0, 1]} \left[(1+\lambda_t)\cdot v_t\cdot F_{conv}(b)- (\lambda_t + \mu_t) \cdot b \cdot F_{conv}(b)\right].$
>
> Regarding the analysis of the regret bound, in Feng et al.'s paper, they point out that the combination of budget and ROI constraints is intuitively straightforward, since the analyses of the algorithms with the two types of constraints respectively are linear in nature. This means one can handle these two constraints separately, and taking the linear combination, the regret under both constraints is still $O(\sqrt{T})$. Overall, the adaptation would be somewhat repetitive technically (with respect to both our analysis of the case with ROI constraints, and Feng et al.'s work [1]), which we wanted to avoid because of limited space; rather, we chose to focus on the additional challenges posed by non-truthful auction mechanisms. We will add a more detailed discussion in the paper.
>
> [1] Zhe Feng, Swati Padmanabhan, and Di Wang. Online Bidding Algorithms for Return-on-Spend Constrained Advertisers.
>
> [2] Gagan Aggarwal, Giannis Fikioris, and Mingfei Zhao. No-regret algorithms in non-truthful auctions with budget and ROI constraints.

---

> > ### Comment · Reviewer_jSPN · 2025-08-01
> >
> > Thanks for you response.

---

> > > ### Author Response · Authors · 2025-08-06
> > >
> > > Thank you again for your time and effort spent reviewing the paper.  We just wanted to quickly follow up and see if any of your concerns still remains.  If it helps, we are more than happy to discuss any outstanding issue and try our best to explain anything that is still unclear.

---

### Decision · Program_Chairs · 2025-09-17

**Decision:**

Accept (poster)

**Comment:**

This paper focuses on the design and analysis of autobidding algorithms for online first-price auctions. More specifically, the focus is on the problem of bidding repeatedly under a certain Return Over Investment constraint. This is motivated by the fact that these kinds of auctions and of constraints are common practice in online advertising auctions. The main challenge in this problem comes from the fact that the opposing bid distribution, and the distribution of the agents' value are unknown and need to be learned online. They show their algorithm can guarantee a regret upper bounded by sqrt(T) under full-information feedback and T^2/3 under bandit feedback.

All reviewers agree that the problem is interesting and the results are clean.
One main concern about this paper is about the lack of experimental results. Even if there is no previously known algorithms for the generalization considered in this paper, experimental results would have been useful (e.g., how much stronger is the stronger benchmark considered in this work with respect to previous ones? If not much, then it would have sense to compare proposed algorithms with algorithms for weaker benchmarks)

Nevertheless. mainly due to the theoretical nature of the paper, these concerns does not outweigh the pros of this paper, and thus we propose to accept this paper.